# A Stress Test for Robustness of Photo Response Nonuniformity (Camera Sensor Fingerprint) Identification on Smartphones

**DOI:** 10.3390/s23073462

**Published:** 2023-03-25

**Authors:** Fernando Martín-Rodríguez, Fernando Isasi-de-Vicente, Mónica Fernández-Barciela

**Affiliations:** Atlanttic Research Center for Telecommunication Technologies, University of Vigo, C/Maxwell S/N (Ciudad Universitaria), 36310 Vigo, Spain; fmartin@uvigo.es (F.M.-R.); fisasi@uvigo.es (F.I.-d.-V.)

**Keywords:** image forensics, camera identification, fingerprint, forgery, PRNU

## Abstract

In the field of forensic imaging, it is important to be able to extract a camera fingerprint from one or a small set of images known to have been taken by the same camera (or image sensor). Note that we are using the word fingerprint because it is a piece of information extracted from images that can be used to identify an individual source camera. This technique is very important for certain security and digital forensic situations. Camera fingerprint is based on a certain kind of random noise present in all image sensors that is due to manufacturing imperfections and is, thus, unique and impossible to avoid. Photo response nonuniformity (PRNU) has become the most widely used method for source camera identification (SCI). In this paper, a set of attacks is designed and applied to a PRNU-based SCI system, and the success of each method is systematically assessed both in the case of still images and in the case of video. An attack method is defined as any processing that minimally alters image quality and is designed to fool PRNU detectors or, in general, any camera fingerprint detector. The success of an attack is assessed as the increment in the error rate of the SCI system. The PRNU-based SCI system was taken from an outstanding reference that is publicly available. Among the results of this work, the following are remarkable: the use of a systematic and extensive procedure to test SCI methods, very thorough testing of PRNU with more than 2000 test images, and the finding of some very effective attacks on PRNU-based SCI.

## 1. Introduction

SCI systems can be very useful for security and digital forensics applications. For example, they can be used to determine if some illegally distributed images come from a particular device. This can be a key point in some criminal investigations. Another application, in the field of security, is authenticating source cameras in surveillance systems, i.e., detecting attempts to impersonate security cameras with false signals [1].

SCI can be implemented on the basis of camera fingerprints that are truly sensor fingerprints. These fingerprints consist of images computed from the intrinsic sensor noise. This phenomenon comes from the unavoidable imperfections present in electronic imaging sensors due to their manufacturing. These imperfections derive into a multiplicative noise that can be modeled in the following equation [2,3,4]:
(1)$$Im_{out}=\left(I_{ones}+Noise_{cam}\right)\cdot Im_{in}+Noise_{add},$$
where *Im_in_* is the true (or pristine) image presented to the camera, i.e., the incident light intensity, I_ones_ is a matrix of ones, and *Noise_cam_* is the “sensor noise”. The center dot is an element-wise matrix product. *Noise_add_* is additive noise from other sources; normally, this term is assumed to be additive white Gaussian noise (AWGN). *Noise_cam_* is a kind of noise due to small differences in light sensitivity between different pixels. Equation (1) expresses the fact that some pixels get bigger values than they should at positions where *Noise_cam_* is positive. Alternatively, in pixels where *Noise_cam_* is negative, image values will be lower than they should be. As stated before, this phenomenon is present in all sensors. Noise pattern (*Noise_cam_*) is not altered during sensor life, and it is different between different chips, even between those manufactured in the same series. Thus, *Noise_cam_* can be used as a sensor fingerprint.

SCI (source camera identification) methods are based on somehow estimating *Noise_cam_* (most times this is achieved by removing noise and then obtaining it as a residue), and comparing results from different images. An attacking method would be any processing over *Im_out_* that to some extent accomplishes the following two conditions: first, making it more difficult to estimate *Noise_cam_* from the processed image; second (important condition), not affecting the visual appearance of *Im_out_* to a great extent. Similar to the importance of cryptanalysis to the cybersecurity world, the study on camera fingerprint attacks is important to image forensics because it is the only way to demonstrate the robustness of proposed extraction and comparison methods.

PRNU estimation has become the “de facto” standard for camera fingerprints. PRNU stands for photo response nonuniformity and it is used to refer to sensor noise as modeled by Equation (1). In [5], an extensive list of SCI methods was compared, and PRNU estimation and comparison was acknowledged as the most used method and the most present in the literature. There are other techniques such as defects due to lens [6], but the authors recognized it as only effective for distinguishing camera models, not individual devices. In [7], Freire-Obregón et al. used a more general approach for the model recognition problem using convolutional neural networks (CNNs). Even sensor dust patterns (in cameras with removable optics such as DSLR) have been used as a feature for SCI (Dirik et al.) [8].

Examples of studies on PRNU-based SCI exist in the literature [1,2,3,4]. The study by Goljan et al. [2] can be seen as a classical reference in this field, in addition to a previous study [9] on the same theme. Other examples include Gisolf et al.’s study [10] with a focus on improving PRNU extraction time without losing accuracy. Moreover, in [11], Goljan studied the “false acceptance probability”, focusing on methods for the comparison of extracted PRNU patterns. In [12], Gupta and Tiwari used preprocessing to improve methods for PRNU extraction. In [13], Uhl and Höller applied PRNU to iris sensors, linking image forensics to biometrics. In [14], Pérez-González and Fernández Menduíña studied information leakage in PRNU computation (i.e., the influence of the content of original images used in the computation), as well as methods for minimizing this leakage. In [15], Bondi et al. studied the issue of PRNU pattern compression for efficient storage and transmission. In [16], Masciopinto and Pérez-González thoroughly investigated the PRNU model and concluded that a revision is needed. In [17], the same authors proposed a new generalized model that takes into account sensor nonlinearities. Nevertheless, PRNU continues to be used in practice, neglecting the aforementioned details. In [18], Debiasi and Uhl studied the use of PRNU for identifying biometric sensors, considering the technological basis of these sensors to be the same as that of the image sensors. Similarly, in [13], Uhl and Höller studied PRNU-based identification in iris sensors. Furthermore, in [19], Gou et al. applied the SCI concept to scanners so as to distinguish scanned from photographed images; the authors of this paper used statistical techniques to characterize sensor noise, thereby obtaining patterns featuring a combination of PRNU and other sensor noise, called dark signal nonuniformity (DSNU).

PRNU has also been used to detect image forgery attempts (image editions with potential illegal purposes). In [20], Chierchia et al. used a Bayesian method after PRNU computation. In [21], Rathgeb et al. focused on detecting the retouching of facial photos.

There are fewer examples of using PRNU-based SCI for videos, a survey of which can be found in [22]. For example, in [23], Al-Athamneh et al. simplified this application by using only the green channel to compute the PRNU, due to the greater importance attached by image sensor designers to the green channel (double the number of samples from the red and blue channels); the authors also claimed that the green channel has stronger noise. In [24], Altinisik and Sencar studied the problem of camera attribution when the video is stabilized through digital processing (typically on-camera processing). Lastly, in [25], Amerini et al. studied the problem of source identification for videos posted on social networks.

The literature contains studies on “counter-forensic” measures (such as those presented in this paper). For example, in [26], Goljan et al. studied methods for defending against attacks on PRNU. In this publication, the authors focused on detecting forging attacks (planting a false PRNU on an image), without erasing or randomly altering the PRNU. In [27], García-Villalba et al. described an anti-forensic method. Their approach was based on wavelet transform, similar to the methods tested here. The authors of [27] did not perform a thorough test of a significant pool of methods unlike this paper. Moreover, they used 50 images per camera, a much smaller quantity than that used herein (more than 200 images per camera and more than 2000 in total). In [28], an open-source project (PRNU Decompare) was presented with software capable of deleting a PRNU pattern or even planting a false one on a given image. Nevertheless, we found no associated paper or report, nor detailed data from systematic tests. In [29], Goljan and Friddrich developed a methodology based on digital communications, considering the image as a disturbing channel for a PRNU signal, to perform SCI against cropping and/or scaling. Nevertheless, simple cropping or scaling does not represent an attack on itslef. It would indeed be interesting to test the proposed approach from [29] against the attacking methods described in this paper. This is left for future work. In [30], Rosenfeld and Sencar performed a study on the robustness of PRNU-based methods, similarly to this paper. However, they only tested common processes: standard processing operations performed by nontechnical people using standard photo editors or open-source software. Thus, we extend this research, using image processing programming to implement real attacks to obfuscate PRNU. In [31], Goljan and Friddrich defined an invariant transform domain that is allegedly able to work against geometrical transformations. Not having access to this software, we cannot check its resiliency against our methods involving operations that are “more than merely geometric transformations” including filtering, denoising, resampling, and noise addition. In [32], Gloe et al. studied a possible attack on a PRNU-based SCI system. A disadvantage is that their attack needs a set of images from the same camera to be effectively applied. We deal with attacks that can be applied to a single image with no access to the source camera. In [33], Dirik et al. studied a special attack method called “seam-carving”; after testing with 50 images per camera (six cameras), the authors found that the effectiveness of the attack varied depending on the camera. They propose customizing attacks depending on the given source camera. In [34], Karaküçük and Dirik evaluated an attack based on noise removal. They alleged that their method could outperform other noise reduction techniques. Without access to this software, we cannot test it; however, this is worth exploring in the future. The study by Entrieri and Kirchner [35] proposed another attack method called patch-based desynchronization. Testing with a total of 500 images, the results were similar to those we obtained with the most successful methods.

In the remainder of this paper, some attacking methods are designed and tested against PRNU-based SCI, enabling insight into SCI resilience and the determination of the most effective methods.

## 2. Materials and Methods

### 2.1. PRNU or Camera Fingerprint

As stated before, PRNU features are extracted from images to accomplish SCI (source camera identification), representing the fingerprint of the camera sensor.

PRNU estimation is always based on some kind of image denoising filter to estimate Im_in_ in Equation (1) (the pristine image). Having a set of images (known to be captured by the same camera), a PRNU pattern can be estimated. A (less exact) PRNU can also be estimated from any image from an unknown source. This new PRNU can be compared to the previously computed PRNU patterns from all possible cameras. When comparison yields a high positive value (PRNU patterns are similar), this means that it is probable that the source camera is the same. For legal applications (e.g., prosecution of child pornographers), despite recent advances in this field, it is very difficult to come to a legally irrefutable result to be used in court.

Note that this process can be performed for any kind of camera, including smartphones. A PRNU pattern can be extracted from any set of images: from one image to an infinite number of them. Pattern quality is greater for more images. Moreover, patterns are better extracted from flat, almost color-constant, images. If it is possible to choose source images (if the camera is available), a collection of flat images can be obtained.

The denoising process used to compute PRNU can vary. For example, in [36] a very simple solution was used: a 3 × 3 median filter. In other examples, the classical Wiener denoising filter [37] was used. The methods proposed in [38,39] were chosen for PRNU computation in this study, where the authors proposed a Wiener-like filter on the wavelet transform domain. Wavelet transform (WVT) [40] is used instead of Fourier transform in the Wiener formula. The classical Wiener equation ($H=\frac{Image}{Image+Noise}$), is normally used in the Fourier domain. To implement this, a certain assumption on noise has to be made. In this case, constant noise power in the WVT domain is assumed.

With any of the previous denoising techniques, the residual *W* is computed as
(2)W=Im−denoise(Im).

Assuming that *W* is equal to the product *Noise_cam_Im_in_*, the additive noise, *Noise_add_* is neglected in Equation (1); assuming also that *Im_in_ = denoise(Im)*, the pattern (or sensor fingerprint) is computed as the weighted average in Equation (3), where N images are available for PRNU computation. Of course, for greater N, a better estimate of PRNU can be computed.
(3)$$F=\frac{\sum_{n=1}^{N}W_{}^{n}Im_{in}^{n}}{\sum_{n=1}^{N}{\left(Im_{in}^{n}\right)}^{2}}.$$

For the identification step, it is important to use a good comparison method to detect similarities between the different extracted patterns. According to [38,39], the best selection is the peak to correlation energy (PCE), which has demonstrated good performance in PRNU comparison [41]. See general schemes for PRNU computation and PCE-based SCI in Figure 1.

PCE is obtained from the complete cross-correlation between two signals (taking into account all possible delays). Defining correlation energy simply as the squared value of correlation, PCE is the quotient of maximum energy divided by the mean energy of the whole correlation. For this mean energy computation, the peak value and its surroundings are EXCLUDED.

### 2.2. Attacking Methods

As stated before, this study aims to test the robustness of PRNU-based SCI. The focus is on methods designed to make SCI more difficult. See Figure 2 for a graphical explanation on attack concept. Methods are classified into four categories, which are described in this subsection. The various methods are described in the subsequent subsections.

The first approach is to use methods based on noise addition (or noise modification). This consists of randomizing the least significant components (bits) (where noise is not added or reduced but merely modified). This is considered as the first approach, because fingerprints are based on noise, and modifying the noise might work. Planting information in the least significant bits is a classical watermarking method. PRNU can be understood as an unavoidable watermark inserted by the camera sensor. Thus, overwriting the least significant bits (LSBs) can create a random watermark, possibly overwriting the PRNU effect.

The second idea is to use geometric distortions. These distortions, e.g., pixel scrambling and/or rotating and de-rotating the image (with a slight angle error), have previously shown success in erasing watermarks. Accordingly, these methods can also be used in PRNU.

A more advanced idea is noise reduction. If image noise is reduced, the fingerprint will also be erased (at least to some extent). Note that PRNU computation methods rely on removing noise first. Thus, removing noise completely would ideally erase PRNU.

Lastly, the most elaborate methods are combined methods, to take advantage of their complementary strengths. As their name indicates, these methods are constructed by cascading two or more of the previous ones, which may be from the same category or not.

#### 2.2.1. Noise Addition (or Modification)

Two approaches are implemented. First, there is the classical approach, where **n** least significant bits in the image (pixel) domain are randomly modified.

The second approach is to adding noise to the discrete cosine transform (DCT) coefficients. In this case, the Watson matrix from the JPEG standard [42] is used to determine the allowed noise quantity. Here, noise is created as a uniform random variable with a different interval (variance) in each DCT position.

#### 2.2.2. Geometric Distortions

Three different approaches are implemented.

The first is based on scrambling pixels, i.e., moving them to a nearby, random position. A maximum radius, **r**, is defined to maintain the process. Gaussian distributed random numbers, N(0,1), are generated and used for computing displacement vectors that move pixels from their original positions. The displacement vector modulus is always less than radius **r**.

The second option is rotating and de-rotating. First, the image is rotated by a significant angle (for example α=15°). Pixel bicubic interpolation is forced; this method produces new pixel values reinforcing PRNU distortion. The image is de-rotated, i.e., rotated again by an angle of −α+β, where β is a small error, e.g., 0.50°. Bicubic interpolation is forced again. This operation produces some artefacts on image corners that can be avoided by simple processing techniques.

The third approach is scaling and descaling. In this case, the image is upscaled by a significant factor (say **sf** = 3). Lanczos3 interpolation [43] is forced. Then, the first line and the first column are erased, and the image is downscaled to its original size, forcing a nonuniform sampling and using Lanczos2 [43] interpolation. The use of different interpolation techniques seeks to create new pixel values that are visually feasible but that obfuscate the PRNU fingerprint. Notice that PRNU is merely noise; thus, it is a weak signal.

#### 2.2.3. Noise Reduction

Two approaches are implemented.

First, the classical Wiener filter is used for de-noising. The Wiener filter is a statistically defined adaptive filter that can reduce noise and distortion in any signal. For an image, the Wiener filter is defined by a frequency response (*H*) according to the equation $H=\frac{Image}{Image+Noise}$. To apply this technique, it is necessary to estimate a value for noise (spectrum or Fourier transform of noise). Typically, white noise is assumed (constant spectrum), and its spectral energy is estimated from the local variance of the image.

The second method is a Wiener implementation in the wavelet transform domain inspired by the PRNU extraction method [38]. This means using the same Wiener equation with wavelet transforms in place for the original Fourier ones.

#### 2.2.4. Combined Methods

As can be easily deduced, combined methods are those constructed by cascading two or more of the previously described methods. These combinations were inspired by our first tests on individual methods, aiming to combine the complementary strengths of the more successful ones. The following combinations are implemented and tested:
Combination of simple noise addition and geometric techniques, without Wiener or other noise reduction (**n** = 3, **r** = 2, α=10°, β=0.50°, **sf** = 3).Wiener filtering, followed by rotation and de-rotation (α=10°, β=0.50°), and then a “deblurring” method for improving image quality, namely, the Lucy–Richardson deconvolution filter [44].


### 2.3. Design of Tests

#### 2.3.1. Testing with Still Images

The first tests were performed using images from the “Dresden Image Database” [45]. In this case, not many images are provided for each device (camera/sensor), i.e., a total of 36 images from six different cameras, with all photos in JPEG format generated by each device with NO processing. Results were not very significant with this small sample, but this was useful to gain insight into the issue.

Secondly, the VISION dataset [46] was used. In this case, six different devices were again selected, but the number of images was much greater with a total of 2057 images from all cameras and 281 images from the less represented one.

As there were different image resolutions, images were preprocessed simply by cropping a centered sub-image of a fixed square size: 2048 × 2048 for the Dresden Images and 1024 × 1024 for the VISION dataset. Images from the VISION dataset were all taken by smartphones. Results in the next section were derived from this dataset as it was the most representative, while also considering that smartphones are currently the most used image-capturing devices.

A detail of some importance is that images may have been taken with a 90° rotated sensor (photos in a vertical format: with the height greater than the width); in this case, preprocessing included rotating them by −90° such that they were always in horizontal format. In addition to testing PRNU-based SCI robustness, these tests also detect which kind of cameras are better (or worse) for SCI. The “Dresden Image Database” consists of images from commercial compact cameras (not smartphones); despite the smaller database size, we can infer results from comparing both datasets. Source camera identification was more effective (less error rate) in the case of smartphones. An explanation is that commercial cameras generally use sensors of higher quality with smaller amounts of PRNU noise.

Attacking algorithms were implemented in MATLAB [47]. Tools were designed to perform tests with the condition that images from each camera were saved in a separate directory. Therefore, adding a new camera to the test is very easy.

For each camera, the first **n_t_** images (**n_t_** = 20 for the VISION database, **n_t_** = 3 for the Dresden database) were used to create a PRNU pattern using the tools in [39].

Next, a confusion matrix for all cameras was computed, understood as the truth table resulting from identification of the source camera for all images not used for computing the PRNU. The i-th row of the matrix accounts was computed when using original images from the i-th camera. The coefficient at i-th row and j-th column is the number of images from the i-th camera classified as coming from the j-th camera. The matrix conveys extensive performance information. Successful SCI recognitions add to the coefficients in the main diagonal. Moreover, the SCI error rate is the sum of coefficients not in the main diagonal divided by the matrix total sum. The confusion matrix is first computed without any attack applied to assess the PRNU-based SCI performance.

Then, each of the attacks defined in Section 2.2 was applied, and the confusion matrix was recomputed, obtaining a new value for the error rate. The change, prospectively rising, in this error rate is the main parameter for evaluating attack performance. Results are presented in the next section.

The SNR (signal-to-noise ratio) after the attack (comparing original and attacked, or processed, images) was also computed as a means for considering image quality degradation. In the next section, the average SNR for each attacking method is presented. The visual quality of processed images was also checked by human volunteers.

Following this experiment, results are summarized and conclusions are drawn to assess the best attacking methods (Section 3). As a starting point, with no attack, the error rate was 25% for the Dresden dataset and 9% for the VISION dataset, confirming that PRNU-based SCI is easier for smartphones.

#### 2.3.2. Testing with Videos

To end our set of experiments, we attempted PRNU-based SCI for video streams. PRNU is equally valid for all frames captured in a video. Typically, a video consists of 25 or 30 frames per second. Cameras usually reduce individual frame resolution when capturing a video: 720 × 576 represents standard definition (SD), 1280 × 720 represents enhanced definition (ED), and 1920 × 1080 represents high definition (HD, or sometimes full HD). There currently exist higher definitions such as 3840 × 2160 (4K) or even 7680 × 4320 (8K), whereby even 4K features < 8 Mpixels per frame.

The question in video SCI is how to combine PRNU information from all frames in a video (assuming that we do not have an edited video with joint fragments from different cameras). We studied video SCI in a previous work public report [48]. In that study, three different approaches were explored.

The first approach consisted of making an independent recognition of each frame, i.e., computing the PCE value for all possible source cameras and selecting the maximum value, which could be considered a vote for a particular camera. The most voted camera reflected the final decision.

The second method was to compute a prorated PRNU pattern for each video to be classified (applying Equation (3)). This pattern was compared (correlated) with the PRNU pattern extracted for each camera.

The final method was to compute a PCE vector (PCE values for all candidate cameras) for each frame. Averaging all vectors, the maximum component reflected the final decision.

After our tests, the best method was that involving voting. See a graphical scheme of this method in Figure 3. In all methods, it was necessary to sample the videos, i.e., not working with all frames. Using all frames would be extremely time-consuming. A sampling period (N) was defined. If N = 10, one of each N = 10 frames was used in the calculations. After tests with different values of N, N = 15 was selected as the one that produced the best results.

Note that these methods make no assumptions about the type of video coding used, and no methods are dependent on the coding format for the selected frame. In other words, some methods use the type of coded frame for frame selection [49,50,51] (in MPEG, frames can be coded as I, P, or B images).

Lastly, an extra option was used to improve SCI results, based on [52], where Taspinar et al. proposed to average the video frames before processing them. In this case, the sample is modified such that, instead of choosing one image out of every N, the mean of N images is calculated. This can be performed in the training phase (calculation of camera patterns) and/or in the execution phase (recognition). The strategy can be implemented for any method, and it is also independent of coding.

The theoretical basis is that, if all the images correspond to Equation (1), when averaging, the result is
(4)$$\bar{Im_{out}}=\left(I_{ones}+Noise_{cam}\right).\bar{Im_{in}}+\frac{\sum Noise_{add}}{N}.$$

Assuming *Noise_cam_* as constant, it represents a common factor in the summation. In contrast, *Noise_add_* differs for each image and is averaged. Therefore, an image with the same usual relationship with the PRNU pattern can be obtained, albeit with averaged frames, but less affected by thermal noise; noise averaging produces a new noise with power divided by N.

Testing with videos from the VISION database, applying SCI based on PRNU using the majority method and N = 15, the results presented below were obtained. The four sampling possibilities were tested. The conclusion is that the results were good, better than for still images; when applying averaging in the execution phase, the results improved.

In Table 1, the aforementioned conclusions for video SCI arise again. First, the results are better than in the still images equivalent. This is reasonable because, in a video, there are many individual frames to test, making it much less probable that errors become a majority. Second, when averaging, results improved only in the running/test stage (not in training). The reason for this could be that reducing additive noise in the recognition stage facilitates classification.

On this basis (0.71% error rate for videos), the most successful attack for still images was applied, yielding the results in Section 3.2.

## 3. Results

### 3.1. Testing with Still Images

First, tests were conducted using the “Dresden Image Database” [45], selecting images from six cameras: Canon Ixus 70 (two cameras of this model), Casio Ex150 (also two instances), and Kodak M1063 (two instances). Secondly, a more thorough test was performed with the VISION dataset [46], selecting again six devices (smartphones), Samsung Galaxy Mini, Apple iPhone 4s, Apple iPhone 5c, Apple iPhone 6, Huawei P9, and LG D290 but with a much greater number of images.

As proposed in the previous section, before any obfuscation process, error rates were 25% for the Dresden dataset and 9% for the VISION dataset. Thus, error rates and other data are, from now on, always described for the VISION dataset.

All attacks increased the error rates. Note that, with six cameras, a random identification algorithm should yield a 16.67% correct identification rate (83.33% error rate). This means that an attack able to produce an error rate greater than this value is a “fully successful” attack.

The error rate is measured twice for each method. First, the training (computation of PRNU patterns) is performed with original (non-attacked) images. Second, training is performed with attacked images. Note that forensic scientists may or may not have access to the suspicious camera. PRNU is often used for clustering [4], aimed at determining whether some images come from the same camera despite no access to it. In our tests, the error rate was always computed with images different from training ones.

The test results are summarized in Table 2, where each method was assigned a “key letter” in alphabetical order, the first method is called **A**. There are two columns related to image quality, which is important because an attack is not useful if it is very noticeable (if image quality is severely damaged). Final image quality is assessed in two ways: subjective (human volunteers) and objective (SNR computation). For some attacks, such as rotating or pixel scrambling, subjective quality can be high but SNR is low because pixels are moved, since SNR is computed by comparing pixels at the same position. Subjective quality was checked with the opinions of seven human volunteers. Here the most repeated opinion is stated, plus some remarkable comments, if any.

Generally speaking, attack success (error rate) was higher when training was conducted with non-attacked images. Perhaps, this is because, when PRNU is computed after being altered by an attack, it contains these alterations and is, thus, more difficult to obfuscate. For example, with the rotating method (method **D**), the error rate went down from 78% to 10% when training with attacked (obfuscated) images. The detected noise pattern is probably rotated by the attack and PRNU robustness is much higher when training with rotated noise. It is remarkable to acknowledge that PRNU is more resilient than considered from the first point of view; PRNU is noise and any decision based on noise seems prone to frequent errors. No method could reach the full success ratio of 83.33%. Some methods were indeed able to come very close when training was conducted with original images. Rates marked in bold can be considered successful attacks as they very significantly reduced the reliability of SCI.

Table 2 provides is a column with the mean execution time for each attack. It can be seen that MATLAB implementation was not efficient. Fast attack implementation was not the main purpose of this work as the focus was on effectiveness. Tests were run on an Intel I5-4590S CPU (3.00 GHz), with 16.00 GB RAM.

In Figure 4, a graphic example is shown. The original image (a) came from a Canon Powershot camera, not from any of the datasets. The image portion dimensions are 2048 × 2048. This image was processed (attacked) using method **I**: Wiener filtering cascaded with rotation and a de-blurring filter, which was one of the more successful ones. Then, PRNU patterns from both images were computed. Note that, in ordinary tests, more than one image is used to get the PRNU pattern and, then this pattern is correlated to those extracted from input images with the aim of guessing the source camera. Using only one source image, PRNU computation is not good for extremely valued pixels (underexposed or overexposed points in the original image); this is translated into dark points seen on the PRNU pattern. In this example, the change between extracted patterns is a more important issue. This kind of pattern is worth showing because it is merely amplified noise consisting of real-valued matrices of the same size as the original image, with values ranging from −16 to 16. In this case, a truncated version of the matrix absolute value is presented. Interpretation simply involves identifying brighter pixels where a significant value for PRNU can be extracted. It can be seen that, for the original image, the algorithm only failed to compute the pattern in some white (perhaps overexposed) pixels. For the processed image, computing the PRNU pattern became much more difficult, whereby dark values were obtained all over the image.

For illustration, an example application of method **H** is presented in Figure 5. Here, the ROI dimensions are 1024 × 1024. The original and obfuscated images are presented. Regarding image quality, the only drawback of this method was the presence of some artefacts on image corners, e.g., the bottom left corner of the attacked image magnified in Figure 5c,d. These artefacts are due to the error in de-rotation (0.50°). In such a process, some pixels are not defined because of the image going outside its original frame when rotating and de-rotating. Two artefact correction methods are illustrated here. The first is computing missing pixels via linear extrapolation. After this method, we can still see some artefacts (Figure 5c). The second method involves cropping the image to erase undefined border pixels and, then rescaling the image to its original size. Better results were achieved using this method, as can be seen in Figure 5d.

Data in Table 2 were already obtained with the new artifact correction process applied to all methods including rotation (**D**, **H,** and **I**).

In Figure 6, two confusion matrixes are presented. 6a is the matrix before obfuscating, showing bigger numbers in the main diagonal (correct source camera recognitions). 6b is the result for the same image collection (the VISION dataset) after attack named **H**. Ideally numbers on each row wiould be approximately equal.

More examples of processed images can be found at https://www.flickr.com/photos/189133275@N08/collections/72157715180732621/ (accessed on 22 March 2023).

### 3.2. Testing with Videos

Testing with videos is based on video material from the VISION dataset (video captured by smartphones). Due to problems with video formats and simulation time issues, we only used data from four cameras (four smartphones): Samsung Galaxy S3 Mini, Huawei P9, Samsung Galaxy Tab 3, and Samsung Galaxy S3. These tests used four videos per camera when training and 11 videos per camera when testing. Each of these 84 videos was at least 1 min long (average duration of 70 s), providing a total of 2800 s for training and 3080 s for testing. The frame rate was 30 FPS, providing 62,400 individual images for training and 92,400 images for testing.

The testing classification with this material before the attack yields the results in Table 1 [48], where it can be seen that, with the best parameters, the classification error is as low as 0.71%. As commented before, the use of many frames per video makes classification errors much less probable.

To test the robustness of this SCI scheme, the attack method **H** was used, which is a combination of noise alteration in the least significant bits and geometric techniques, with the best performance according to Table 2. In this case, attacking means decoding frames, executing the attack on each frame, and recoding the full video. This attack is applied to the whole image, regardless of the original resolution. When testing, all frames are decimated to enhanced definition (ED: 1280 × 720) which is merely the minimum video resolution for any contemporary camera (ordinary camera, webcam, or smartphone). In this step, it is important to decimate images (using the so-called nearest mode) to get only original pixels from the image, but not those coming from averaging or interpolation methods.

Data from Table 1 were recalculated, yielding new results (Table 3). From these new data, the conclusion is that video SCI, at least using the method defined in [48], is more resilient that its still images counterpart. Nevertheless, attacks strongly obfuscated the SCI process, making it almost unusable. Remember that the SCI method for videos is based on individual frame classifications and voting. Not all images are used, instead sampling one image out of N (N = 15). The average in Table 3 refers to the N original frames extracted from each sample. The behavior changes in this case, presenting a better result when averaging in the two stages.

Other data from this test included the average SNR per obfuscated frame (19 dB) and the fact that MATLAB implementation was extremely time-consuming (81.85 s on average) for obfuscating 1 s of video (30 FPS, 1280 × 720, ED).

## 4. Discussion

### 4.1. General Conclusions and Future Work

The robustness of the currently most used method for camera fingerprint identification (PRNU extraction and PCE comparison) was thoroughly tested; the conclusion is that it has strong resilience against simple attacks such as those based on geometric operations and/or based on altering noise. More elaborated methods based on noise theory such as Wiener filtering were also tested creating better, but still not definite, attacks. The best attacks in our study were created by combining pairs of individual methods that performed well.

As a general conclusion, PRNU’s initial effectiveness and resilience were greater in cameras (Dresden Database) than in smartphones (VISION dataset). A greater image sensor quality leads to worse effectiveness of SCI (source camera identification).

Among the results of this work, the following are remarkable: the development of a systematic procedure to test SCI methods, very thorough testing of PRNU with more than 2000 test images, and, the finding of some very effective attacks on PRNU-based SCI.

SCI robustness was also assessed for videos, drawing the conclusion that, in this case, it is more difficult to obfuscate SCI classification because of the great amount of information present in a video. Nevertheless, obfuscation is possible to a level that makes SCI classification almost useless.

### 4.2. Method Selection

According to the last two columns of Table 2, it can be concluded that methods with rotation are better for obfuscating PRNU.

It can also be concluded that obfuscation is much more effective on forensic systems trained with “non-fooled” images. Training with obfuscated images always leads to smaller errors. For this reason, image clustering based on PRNU similarities remains possible in the presence of simple fooling methods, whereas attribution of “fooled” images to a known camera (even with physical access to it) is more difficult.

According to the test results, method **H** can be considered the most successful attack method, i.e., the combination of simple noise addition and geometric techniques. This is because it produced the largest error with acceptable quality.

In the video tests, only method **H** was tested, yielding the results in Table 3. Other methods would probably yield worse results because **H** was the better obfuscator for individual frames.

### 4.3. Future Work

Future research can include new methods, perhaps taken from some of the publications mentioned in Section 1, to more deeply study the interactions between attack methods and fingerprint computation. This would also allow the development of more robust fingerprint extraction methods.

## Figures and Tables

**Figure 1 sensors-23-03462-f001:**
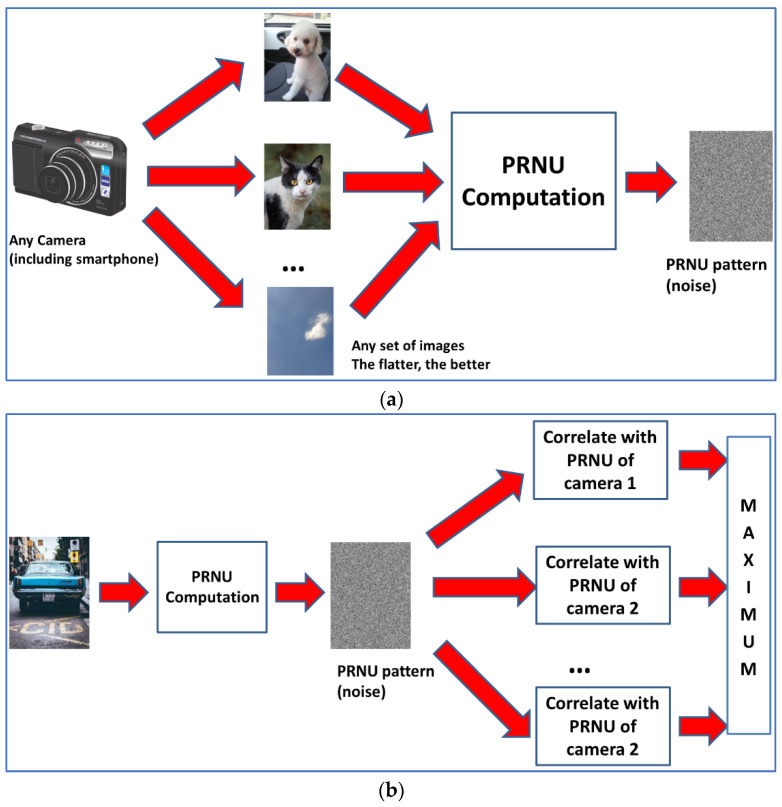
(**a**) Computation of PRNU from a given camera. (**b**) Source camera identification.

**Figure 2 sensors-23-03462-f002:**
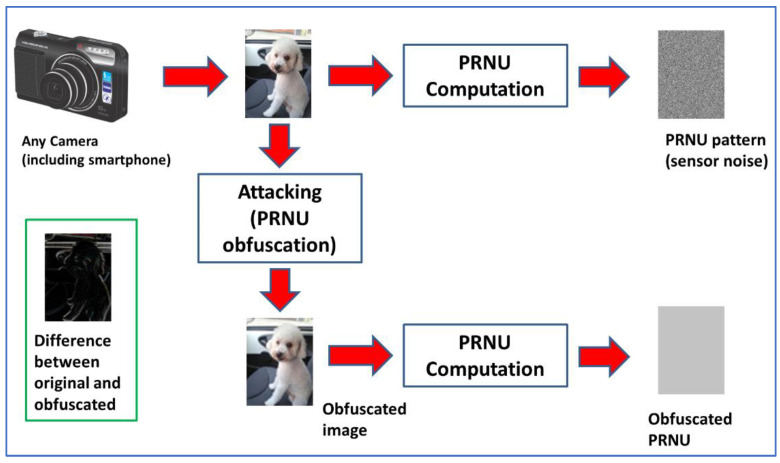
Concept of attack over PRNU-based SCI, graphical depiction.

**Figure 3 sensors-23-03462-f003:**
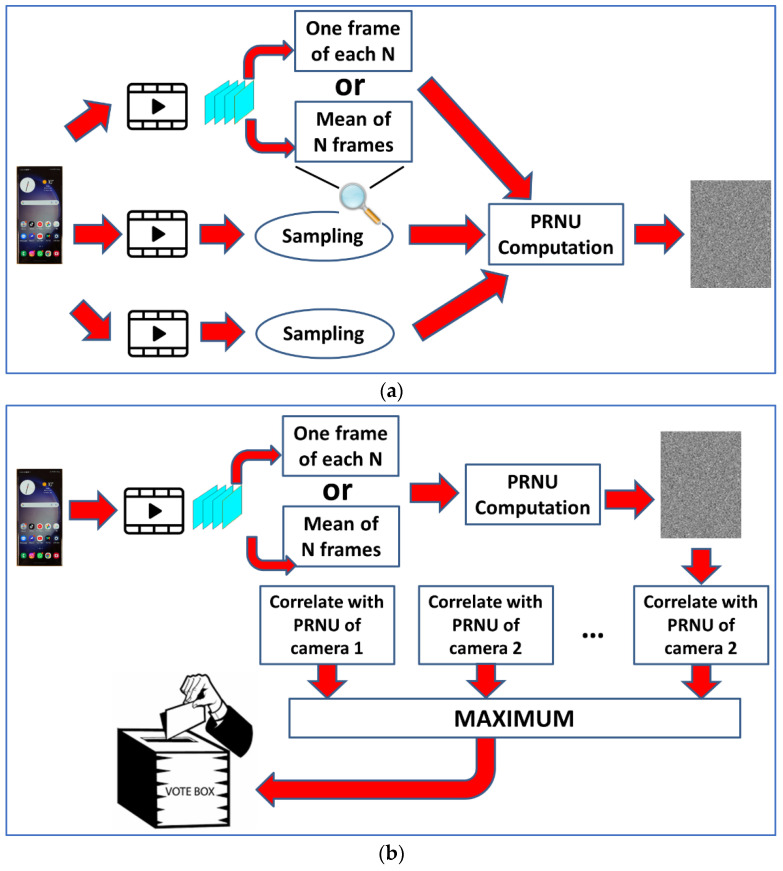
SCI for videos: (**a**) characterization (training); (**b**) recognition.

**Figure 4 sensors-23-03462-f004:**
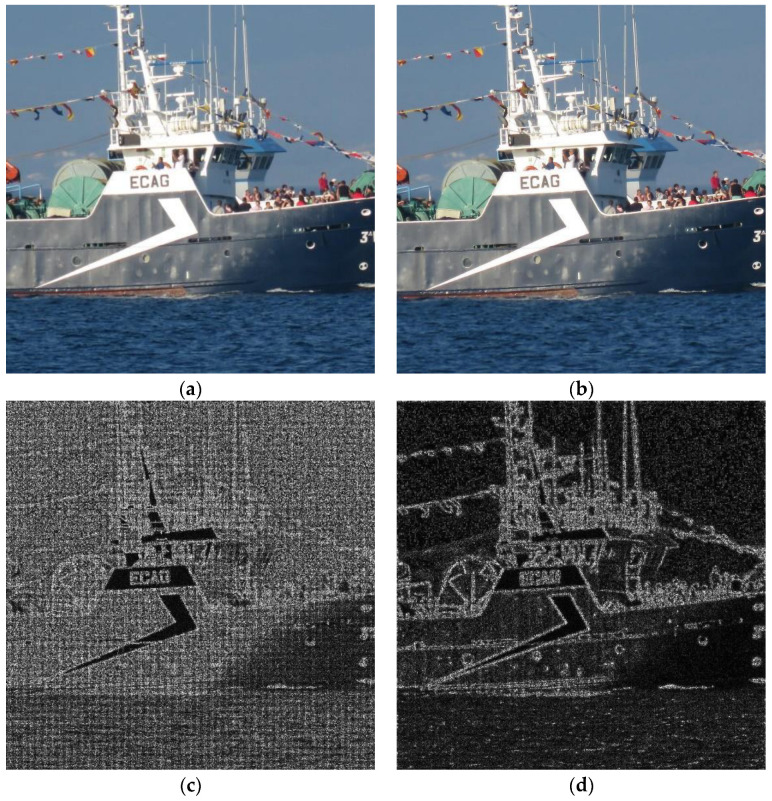
(**a**) Original image; (**b**) image processed using method **I**; (**c**,**d**) noise patterns (PRNU) obtained from each of the above images.

**Figure 5 sensors-23-03462-f005:**
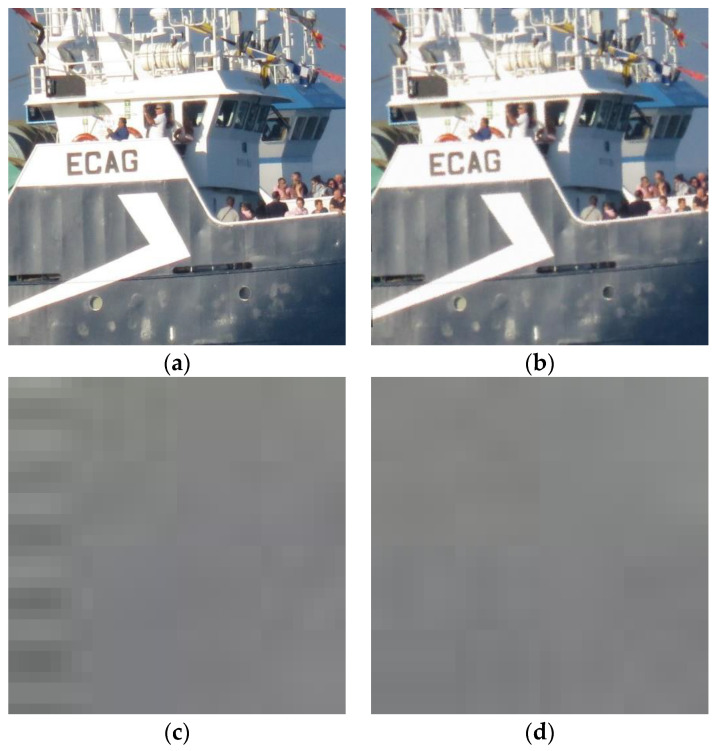
ROI (1024 × 1024) of the original image of Figure 2 (**a**); image processed with method **H** (**b**); corner artefacts corrected with different methods (**c**,**d**).

**Figure 6 sensors-23-03462-f006:**
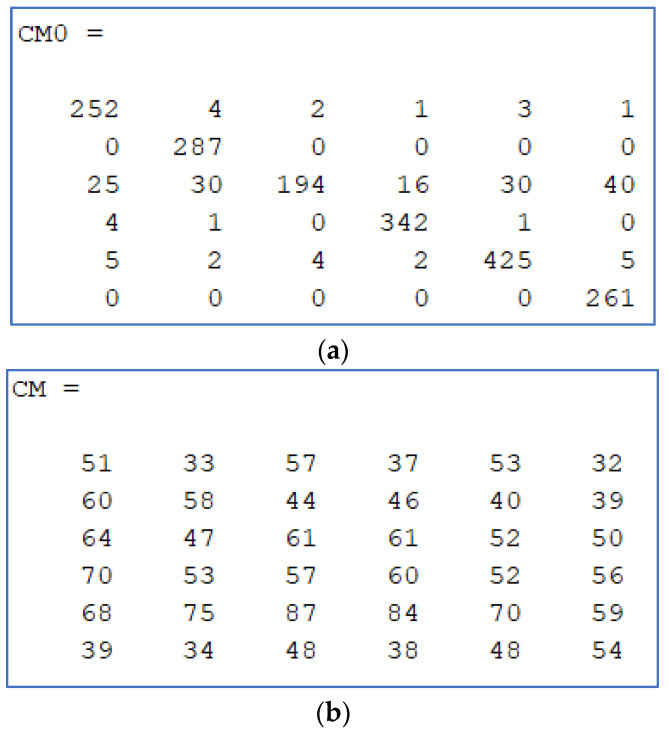
Confusion matrix samples: (**a**) no attack; (**b**) preferred combined method (method **H**).

**Table 1 sensors-23-03462-t001:** SCI on video, test material. VISION dataset (smartphone videos); method: majority, N = 15.

Average in the Training Phase	Average in the Running Phase.	Error (%)
NO	NO	3.55
YES	NO	4.26
NO	YES	**0.71**
YES	YES	2.13

**Table 2 sensors-23-03462-t002:** Summary of test results (original error rate: **9.09**).

Attacking Method:	Key Letter:	Visual Quality:	SNR (av. dB):	Mean Exec. Time (s):	<<Error Rate>>, Non-Fooled Train Set (%).	<<Error Rate>>, Fooled Train Set (%).
Aleatorizing least significant bits (**n** = 3)	**A**	Good, except color degradations (clouds).	38	2.59	9.30	10.48
Introducing noise on DCT coefficients	**B**	Good	49	6.48	9.50	9.55
Scramble randomly pixels (r = 1)	**C**	Good	31	3.24	11.25	11.72
Rotating and de-rotating (A = 10°, a = 0.5°)	**D**	Good, except for artifacts on borders.	22	2.88	**78.16**	9.75
Scaling and de-scaling (sf = 3)	**E**	Good	44	3.36	9.19	9.30
Ordinary Wiener filter	**F**	Good	31	0.25	32.58	23.28
Wavelet transform Wiener filtered and inverted	**G**	Good	41	0.51	10.74	12.75
Combination of simple noise addition and geometric techniques (n = 3, r = 2, A = 10°, a = 0.5°, sf = 3)	**H**	Good, artifacts in some borders, quantification in color degradation areas (sky, clouds).	23	3.68	**81.72 (*)**	47.23
Wiener + rotating/de-rotating + deblurring (Lucy)	**I**	Good	23	2.90	**78.83**	25.29

* Best result (this is the best method for obfuscating PRNU).

**Table 3 sensors-23-03462-t003:** SCI on video, test material. VISION dataset (smartphone videos); method: majority, N = 15.

Average in the Training Phase	Average in the Running Phase	Error (%)
NO	NO	61.90
YES	NO	50.00
NO	YES	59.92
YES	YES	45.24

## Data Availability

All results came from public image databases: Dresden Image Database and Vision dataset.

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
