# Peer review of "A Stress Test for Robustness of Photo Response Nonuniformity (Camera Sensor Fingerprint) Identification on Smartphones"

_sensors, 2023, doi:10.3390/s23073462_

Round 1

Reviewer 1 Report

The authors did experimental work to identify smartphones using PRNU. Different modifications are also applied to the images and videos to check the robustness of existing PRNU-based camera identification schemes. The authors consider many scenarios for testing.

I have a few recommendations and queries as follows:

1.      Introduction/related work section should comprise a discussion of more research articles. The present discussion is not sufficient. Several popular schemes are missing.

2.      I don’t find any scheme or method proposed by the authors. The proposed scheme steps should be defined and also need to demonstrate the motivation for choosing a particular approach.

3.      Many words/sentences are mentioned using double quotes “…..”. It is highly unusual.

4.      I don’t understand the motivation for doing testing only.

5.      The captions of figures and tables should be defined appropriately. Terms like the above and below are not suitable.

6.      Mention the tables and figures in the text using their respective number, not by below or above words.

7.      Table caption should be defined top of the Table. The figure caption should be defined bottom of the figure. Please follow the manuscript format guidelines.

8.      Figure 3 discussion needs to be included in the text.

Author Response

General comments: Authors wish to thank all reviewers for their comments that have been very useful for improving our manuscript.

Article is about camera identification using noise patterns (that are commonly called sensor fingerprints). Sensor fingerprint (or PRNU patterns) can be obtained from normal camera images. May be a smartphone (any smartphone) or any other kind of camera. Article explores robustness of this method for camera identification.

Explanation on changes to the manuscript: entire text has been reviewed for English language correctness, number of bracketed expressions has been reduced and most use of quotation marks has been removed (there was some abuse on this; we newly appreciate the work of reviewers for indicating this).

Explanations have been rephrased at many parts to improve readability. Some graphical schemes were also added to improve comprehension of some key processes as PRNU computation and SCI process for images and videos.

Bibliographic study was extended with new examples and new references. Thanks again to reviewers for their comments.

Please excuse us because, we forgot to activate version control in MS-word for the first revisions. Nevertheless, most important changes are highlighted.

Responses to particular items:

  1. “Introduction/related work section should comprise a discussion of more research articles. The present discussion is not sufficient. Several popular schemes are missing.” Discussion has been extended. We thank reviewer 1, for suggesting this improvement.
  2. “I don’t find any scheme or method proposed by the authors. The proposed scheme steps should be defined and also need to demonstrate the motivation for choosing a particular approach”. Article purpose is testing robustness and searching for efficient attacks. PRNU based SCI is performed in the classical way from publications by M. Goljan et al. All attacks are proposed by authors. In SCI for videos, three authors’ designed methods are described before testing. Some graphical schemes have been added to improve comprehension. Comment is appreciated because it made us clear that we had to explain this better.
  3. “Many words/sentences are mentioned using double quotes“…..”. It is highly unusual.” Most of them have been removed in our first general revision. Thanks.
  4. “I don’t understand the motivation for doing testing only.” PRNU based SCI is well established just now, testing its robustness and thorough testing on smartphones is not so popular. Testing the robustness of a forensic method is important, just as cryptanalysis is for ciphering.
  5. “The captions of figures and tables should be defined appropriately. Terms like the above and below are not suitable.” This issue has been corrected. Thanks to reviewer for this help.
  6. “Mention the tables and figures in the text using their respective number, not by below or above words.” Terms like the above and below are not suitable.” This issue has been corrected. Thanks to reviewer for this help.
  7. “Table caption should be defined top of the Table. The figure caption should be defined bottom of the figure. Please follow the manuscript format guidelines.” This issue has been corrected. Thanks to reviewer for this help.
  8. “Figure 3 discussion needs to be included in the text.” This issue has been corrected. Thanks to reviewer for this help.

Thanks a lot for your comments.

Best regards...

Reviewer 2 Report

1. Give a functional diagram of the developed method. Give the algorithms of the developed method.

2. You claim to recognize fingerprints for a smartphone. For which class of smartphones? What types of sensors and what software should be installed there?

3. The list of attacks in table 2 is far from complete. Give, at least in the text of the article, a detailed classification of attacks on fingerprints.

4. You give the formula in the first paragraph. I consider it incorrect. Also decipher all the symbols.

5. I noticed the decoding of SCI (Source Camera Identification) three times. Why repeat several times? In the title of the article, you give the abbreviation - PRNU. This is incorrect.

6. The author of the article is very fond of quotation marks, in particular “vote”, “new”, “non-uniform sampling”, “Lanczos2”, “attack” and the like. Reduce their number. This is incorrect.

7. Reduce the number of brackets in the text.

8. You say that you recognize fingerprints, but you give a drawing of a ship.... Is this correct?

Author Response

General comments: Authors wish to thank all reviewers for their comments that have been very useful for improving our manuscript.

Article is about camera identification using noise patterns (that are commonly called sensor fingerprints). Sensor fingerprint (or PRNU patterns) can be obtained from normal camera images. May be a smartphone (any smartphone) or any other kind of camera. Article explores robustness of this method for camera identification.

Explanation on changes to the manuscript: entire text has been reviewed for English language correctness, number of bracketed expressions has been reduced and most use of quotation marks has been removed (there was some abuse on this; we newly appreciate the work of reviewers for indicating this).

Explanations have been rephrased at many parts to improve readability. Some graphical schemes were also added to improve comprehension of some key processes as PRNU computation and SCI process for images and videos.

Bibliographic study was extended with new examples and new references. Thanks again to reviewers for their comments.

Please excuse us because, we forgot to activate version control in MS-word for the first revisions. Nevertheless, most important changes are highlighted.

Responses to particular items.

  1. “1. Give a functional diagram of the developed method. Give the algorithms of the developed method.” Functional schemes have been added for better explanation of PRNU based SCI methods. We thank reviewer 2, for suggesting this improvement.
  2. “You claim to recognize fingerprints for a smartphone. For which class of smartphones? What types of sensors and what software should be installed there?” As stated before, a PRNU pattern (commonly called: camera fingerprint) can be computed from any number of camera images (the greater number, the more exact calculation). This can be applied to any digital camera, including ALL smartphones. No software in smartphone/camera is required. Processing is done over the jpeg images copied in a computer.
  3. “The list of attacks in table 2 is far from complete. Give, at least in the text of the article, a detailed classification of attacks on fingerprints.” Explanation of attacks has been improved from the classification part until the end. Another graphical scheme has been added. Attack design is original from authors. Thanks for comment.
  4. “You give the formula in the first paragraph. I consider it incorrect. Also decipher all the symbols.” Explanation on equation 1 has been extended. This equation is present on the first and classic reference for camera identification: https://ieeexplore.ieee.org/document/4451084 (reference 2 on article) and in other research papers as: https://ieeexplore.ieee.org/document/9287451.
  5. “I noticed the decoding of SCI (Source Camera Identification) three times. Why repeat several times? In the title of the article, you give the abbreviation - PRNU. This is incorrect.” This issue has been corrected. In fact, we have taken advantage to rewrite the title searching for a more appealing one. Thanks to reviewer for this help.
  6. “The author of the article is very fond of quotation marks, in particular “vote”, “new”, “non-uniform sampling”, “Lanczos2”,“attack” and the like. Reduce their number. This is incorrect.” This issue has been corrected. Thanks to reviewer for this help.
  7. “Reduce the number of brackets in the text.” This issue has been corrected. Thanks to reviewer for this help.
  8. “You say that you recognize fingerprints, but you give adrawing of a ship.... Is this correct?” It is about the fingerprint of camera (PRNU), not a biometric fingerprint. It can be obtained (or tested) from any camera image (or set of images).

Thanks a lot for your comments.

Best regards...

Reviewer 3 Report

Excellently investigated and recommend for publication

Author Response

General comments: Authors wish to thank all reviewers for their comments that have been very useful for improving our manuscript.

Article is about camera identification using noise patterns (that are commonly called sensor fingerprints). Sensor fingerprint (or PRNU patterns) can be obtained from normal camera images. May be a smartphone (any smartphone) or any other kind of camera. Article explores robustness of this method for camera identification.

Explanation on changes to the manuscript: entire text has been reviewed for English language correctness, number of bracketed expressions has been reduced and most use of quotation marks has been removed (there was some abuse on this; we newly appreciate the work of reviewers for indicating this).

Explanations have been rephrased at many parts to improve readability. Some graphical schemes were also added to improve comprehension of some key processes as PRNU computation and SCI process for images and videos.

Bibliographic study was extended with new examples and new references. Thanks again to reviewers for their comments.

Please excuse us because, we forgot to activate version control in MS-word for the first revisions. Nevertheless, most important changes are highlighted.

Thanks a lot for your comments.

Best regards...

Round 2

Reviewer 1 Report

The authors have revised the manuscript according to the comments. I don't have any further significant comments.

Author Response

General comments: Authors wish to thank all reviewers and editors for their comments that have been very useful for improving our manuscript.

Article is about camera identification using noise patterns (that are commonly called sensor fingerprints). Sensor fingerprint (or PRNU patterns) can be obtained from normal camera images. May be a smartphone (any smartphone) or any other kind of camera. Article explores robustness of this method for camera identification.

Explanation on changes to the manuscript: entire text has been reviewed for English language correctness. Suggestions of academic editor have been taken into account adding information where necessary and moving some paragraphs to more adequate sections.

Reviewer 2 Report

no comments

Author Response

(The authors gave the same response as above.)
